# Bringing atom probe tomography to transmission electron microscopes

Gerald Da Costa [1], Celia Castro [1], Antoine Normand[1], Charly Vaudolon[1], Aidar Zakirov [1], Juan Macchi [1], Mohammed Ilhami [1], Kaveh Edalati[2], François Vurpillot [1] & Williams Lefebvre [1] ✉

For the purpose of enhancing the structural insights within the three-dimensional composition fields revealed by atom probe tomography, correlative microscopy approaches, combining (scanning) transmission electron microscopy with atom probe tomography, have emerged and demonstrated their relevance. To push the boundaries further and facilitate a more comprehensive analysis of nanoscale matter by coupling numerous two- or three-dimensional datasets, there is an increasing interest in combining transmission electron microscopy and atom probe tomography into a unified instrument. This study presents the tangible outcome of an instrumental endeavour aimed at integrating atom probe tomography into a commercial transmission electron microscope. The resulting instrument demonstrates the feasibility of combining in situ 3D reconstructions of composition fields with the detailed structural analysis afforded by transmission electron microscopy. This study shows a promising approach for converging these two important nanoscale microscopy techniques.

Since its invention just over 30 years ago[1,2], Atom Probe Tomography (APT) has emerged as a pivotal tool in the design of nanostructured materials[3,4], addressing the challenges posed by modern metallurgy[5,6], energy storage[7], and answering fundamental questions into geological scenarios of the past[8]. As an inherently 3D characterisation technique, APT provides simultaneously the position and elemental nature of individual atoms with near-atomic spatial resolution. To extend the intrinsic capabilities of APT, correlative approaches combining (scanning) transmission electron microscopy (STEM and TEM), have established a cutting-edge methodology to perform microscopy at a near-atomic scale[9–16]. Meanwhile, an increasing interest has emerged to unify these two microscopy tools in a single instrument[17–20].

In the collective psyche, the microscope remains the indispensable tool of researchers, likely because one vision of research is to explore matter and physical phenomena at scales invisible to the naked eye. Since the advent of the first optical microscopes, microscopy has diversified extensively and continually evolved. Recent developments, leveraging material responses to different types of excitations, enable pushing the boundaries of spatial and temporal resolutions while combining a multiplicity of spectral and structural information. Time-resolved microscopy[21,22], the emergence of in situ and operando microscopy[23,24], advancements in sources, detectors and numerical methods[25], continue to broaden the horizons of microscopy. Nowadays, the most advanced microscopes and methodologies can facilitate the mapping of electrostatic or magnetic fields at the atomic scale[26–28], and in some cases, analytical electron tomography[29,30].

Nonetheless, is there currently a technique capable of reconstructing matter with ultimate precision, atom by atom, unambiguously identifying their nature, while accessing different fields of three-dimensional properties (optical, mechanical, electronic)? Regrettably, not yet. However, a pathway towards the development of such an instrument and towards the concept of Atomic Scale Analytical Tomography[31] has been proposed by combining two of the most powerful microscopy tools: STEM and APT[14]. The design of such an instrument would allow merging the unmatched spatial resolution of

[1]Univ Rouen Normandie, INSA Rouen Normandie, CNRS, Normandie Univ, GPM UMR 6634, F-76000 Rouen, France. [2]WPI, International Institute for Carbon-Neutral Energy Research (WPI-I2CNER), Kyushu University, Fukuoka, Japan. ✉e-mail: williams.lefebvre@univ-rouen.fr

STEM with the atom-by-atom reconstruction capabilities of APT, as demonstrated so far by ex situ correlative microscopy approaches[4,9–12,32] (the term "ex situ" refers here to the use of two distinct instruments), which highlight the allure for combining (S)TEM with APT in a single instrument for high-resolution material analysis.

Though it combines a unique chemical sensitivity (down to parts-per-million) with intrinsic 3D capability, APT suffers from limited spatial positioning precision inherent to the physical principles underlying this technique[33–35]. APT allows collecting typically 10 to 100 million of atoms and to reconstruct volumes in the range of $(50–100) \times (50–100) \times (200–1000)$ nm$^3$. Each specimen must be prepared in the shape of sharp needles, with a curvature radius in the range of 20–100 nm at their apex. After being field evaporated from the specimen surface, by means of ultrashort laser or voltage pulses superimposed to a DC voltage, ions are accelerated across the electrostatic potential gradient facing the specimen in ultra-high vacuum and detected by a position sensitive detector[33–35]. The detection efficiency of the technique may vary from 37% to 80 % according to the instrument setup[33–35]. Volumes are later reconstructed, atom-by-atom, according to geometrical models which mostly require prior knowledge (or hypothesis) of the specimen geometry, before, during, and/or after analysis.

The anisotropic spatial precision of APT (smaller than a fraction of picometre along the direction of analysis but in the sub-nanometre range along the detection plane) is a result of the atomic scale variations of electrostatic field along ion trajectories at the onset of field evaporation[35]. Electron microscopists have hence proposed to overcome this physical limitation by providing knowledge of specimen structure[10,13,32,36,37] or by determining the 2D or 3D distribution of electrostatic field in the vicinity of the polarised specimen[38,39]. Combining these additional data with algorithmic methodologies such as the lattice rectification[40] may be a way to achieve enhanced and isotropic resolution in APT. Nevertheless, another obstacle would remain which is the temporal evolution of the APT specimen's surface along the acquisition sequence. Indeed, the atom-by-atom field evaporation induces a dynamic evolution of the electrostatic environment at the vicinity of the specimen's apex, which results in a dynamic change of geometrical parameters controlling the 3D reconstruction[41] (see the evaporation sequence provided in the supplementary information).

Figure 1 illustrates what in situ correlative microscopy by APT and STEM (in situ refers here to the combination of APT and STEM in a single instrument) could provide if APT was available in a Cs-corrected STEM. The image merges the chronologic analysis of an Al-7wt%Ag alloy started in APT before moving the specimen to a STEM equipped with a spherical aberration corrector (Cs-corrected STEM). Here, an APT analysis has been stopped and the APT needle has been mounted on a double-tilt STEM holder to perform high-resolution observations. The distribution of Ag atoms in the projected portion of APT reconstruction illustrates the high sensitivity of the technique, which provides access to 3D chemical field mapping. Furthermore, the knowledge of the nanostructure is improved here by the atomic resolution and the contrast of atomic numbers provided by the STEM image. The example in Fig. 1 makes it evident that the ability to carry out similar observations repeatedly and easily in a single instrument would represent a major step forward.

Though Walck demonstrated in 1986 the possibility of applying high voltage, up to 3 kV, to atom probe specimens inside a TEM[42], it took several decades for the idea of implementing an Atom Probe in a TEM to arise, given that the techniques, when viewed independently, initially seemed significantly incompatible. APT operates under ultra-high vacuum ($10^{-9}$ Pa compared to $10^{-6}$ Pa in TEM) and cryogenic temperature. Consequently, significant concerns exist about the quality of mass spectra and quantitativity of measurements of an APT placed in a TEM. Furthermore, it requires application of high voltage to the specimen (1–8 kV) and the use of voltage pulses (-20% of the DC

voltage) or ultrashort laser pulses (sub-picosecond range)[33–35]. Such requirements are a priori incompatible with the environment of a TEM. Indeed, the objective pole piece gap offers only a few millimetres to the specimen, where a large magnetic field is applied. Nevertheless, encouraging achievements about the possibility to image field evaporation in TEM[43] even with high resolution[44] were obtained by using moderate voltage values (in the range of 100 V) while approaching a counter electrode from needle shape specimens. An earlier attempt to join APT and electron microscopy was proposed by Larson et al.[45], who used electron pulses (in a scanning electron microscope) instead of electric pulses to produce triggered field evaporation. More recently, Kirchoffer et al.[46] have proposed to mount an electron gun on a modified Cameca local electrode atom probe (LEAP®) and have showcased the possibility to perform electron diffraction on needles in an APT. In 2013, T. Kelly and co-authors have proposed a roadmap to join an ultra-high vacuum STEM and an APT in a single instrument[17]. In the continuity of this idea, efforts have been pursued under the leadership of the Ernst Ruska Centre for Microscopy and Spectroscopy with Electrons in Jülich (Germany) who placed an order to instrument suppliers (Thermofisher in collaboration with Cameca) to build a unique instrument combining an APT with a STEM[19,47].

## Results and discussion

The approach of the present study is significantly different[18]. The aim here is to build an APT that can be easily adapted to commercial (S) TEMs. Our instrumental approach is based on the design of two setups to be added to the microscope. The first setup is a specialised APT-(S)TEM holder, fitting the microscope's goniometer (without any modification). The designed holder enables analysis at cryogenic temperature (measured down to 78 K), application of DC voltage up to 8 kV and of voltage pulses (up to 3 kV) at a maximum frequency of 20 kHz. The second setup is a home-made APT detector to be mounted on the microscope's column, in the alignment of the specialised APT-(S)TEM holder. It is based on the technology presented in the following reference[48]. A schematic diagram of the instrument is displayed in Fig. 2 (an image of the whole APT-(S)TEM instrument is available in the supplementary information).

Before trying to mount an APT detector in a commercial TEM, it was mandatory to establish whether high voltage polarisation (up to several kV) of an APT needle is compatible with simultaneous imaging in TEM. For this purpose, a specimen holder compatible with a large range of TEMs was first designed to enable polarisation up to 8 kV of a sharp needle at room temperature. This was tested with a previous generation of electron microscope. Field evaporation sequences of tungsten needles could be recorded and qualitative mapping of electrostatic field variations in the vicinity of APT needles could be mapped (see the supplementary information). Following these preliminary proofs of concept, the complete APT was built and connected to a commercial JEOL F2 (S)TEM. The microscope offers versatile imaging in TEM and STEM conditions, with a 4D-STEM mode[49,50], which means that for a scanned area of the specimen, a diffraction pattern can be recorded for each pixel. The processing of diffraction datasets can lead to crystal orientation mapping, imaging of crystal defects or multiple field mapping (e.g., elastic strain, electrostatic field)[49,50]. The methodology displayed in Fig. 2 illustrates how processing of recorded diffraction patterns leads to orientation mapping of nanocrystals. Results shown in the following were obtained with this configuration.

In order to first evaluate the quantitativity of the analyses realised by APT in the electron microscope, the same material (a Fe-51.4at% Cr alloy) in the similar thermomechanical condition was also investigated for comparison with a dedicated atom probe (LEAP 5000 XS®), which is a straight atom probe (i.e., without any reflectron, which can be mounted to improve the mass resolution). Respective mass spectra in the region of $Fe^{2+}$ and $Cr^{2+}$ isotopic peaks are overlapped in Fig. 3. Despite a slightly higher level of background noise and lower mass

resolution, all stable isotopes of Fe and Cr can be unambiguously distinguished and selected for 3D elemental mapping. It must be mentioned that the mass spectrum data displayed in Fig. 3 for the APT in JEOL F2 is plotted without any background substraction. The mass resolution at 10% is more strongly affected for the APT inside the TEM than the Full Width at Half Maximum (FWHM) value. The FWHM value demonstrated here, without any hardware correction, allows an easy distinction of peaks in the present application.

Figure 4 showcases some of the opportunities of advanced characterisation offered by in situ correlative analysis by APT and STEM. For a purpose of illustration of the instrument, the Fe-51.4Cr(at%) alloy was processed to reach an ultrafine grain size distribution. It was successively characterised by STEM and APT in the same instrument

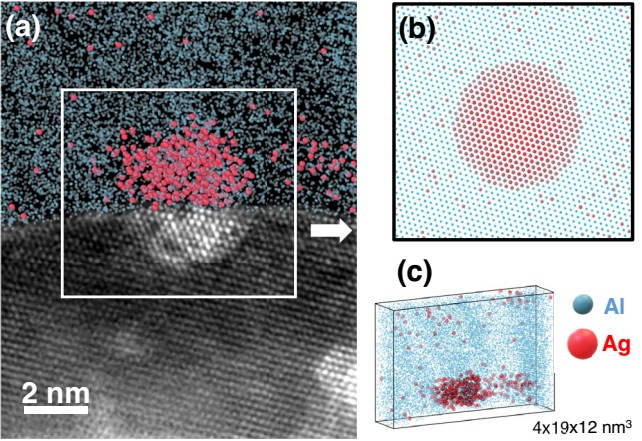

**Fig. 1 | Illustration of an ex situ APT-STEM correlative analysis of an Ag-rich particle partially analysed in APT and STEM successively. a** Superimposition of an APT reconstruction (upper part) which was interrupted to observe the remaining part of the APT specimen with Z-contrast STEM imaging (lower part). **b** Sketch representative of a spatially resolved 2D projection accessible in STEM. **c** Perspective view of the 3D APT reconstruction shown in (**a**).

and multiple data were acquired at various steps of the field evaporation of the specimen (Fig. 4). For the specific heat treatment applied to the alloy, a spinodal decomposition[51–53] occurs, during which waves of concentration fields establish while both their wavelength (in the nanometre range) and amplitude evolve as a function of the annealing time. The potential interaction of spinodal decomposition with the ultrafine grain size is of particular interest. Though 3D composition fields can easily be mapped by APT, the localisation and definition of grain boundaries (GB) is hardly accessible with APT and needs input data from diffraction in 4D-STEM[50].

At stage $t_2$ (Fig. 4a), a 4D-STEM acquisition was performed in order to reveal the grain structure in the region analysed in APT (Fig. 4b). This was done by processing each diffraction pattern obtained in precession mode for each pixel of the image with the methodology presented in ref. 50. In the APT reconstruction recorded in after step $t_3$ (Fig. 4c), the complex entanglement of Fe and Cr-rich regions is revealed. The relation between this entanglement and the location and nature of GBs is done by considering both the map in Fig. 4b and the diffraction patterns associated with grains 1, 2 and 3 crossed during the APT analysis. This additional information reveals that GB1 is a high-angle GB, whereas GB2 is close to a low-angle twisting GB. The effect of GB1 on the composition fields is obvious owing to the presence of Cr-rich region parallel to the GB plane. On the opposite, no clear influence of GB2 on the distribution of Fe and Cr can be revealed. While additional analyses and comparisons to models are necessary to further assess this effect in detail, this illustration underscores the significance of integrating APT and STEM in a single microscope, particularly when structure and chemistry interact in materials at the nanometre scale.

Beyond the possibility of in situ correlative analysis, the dynamic imaging of the field evaporation by TEM is seen as a unique opportunity to access the instantaneous geometry of the APT specimen. Nevertheless, questions remain about the actual TEM resolution in the presence of the high electrostatic field required for field evaporation (in the 10–30 V/nm range). To evaluate this possibility with the setup used in the present work, an evaporation sequence has been recorded (see Supplementary Movie 2.avi and Figure S5 of the supplementary information). The point to point resolution of the JEOL F200 in TEM

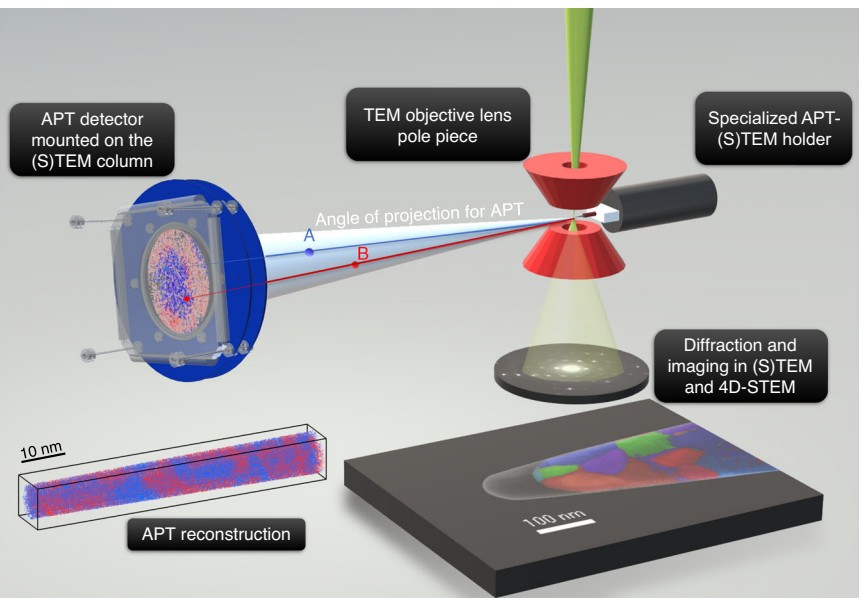

**Fig. 2 | Sketch illustrating the implementation of atom probe tomography in a scanning transmission electron microscope.** The APT specimen is inserted in the objective lens pole piece, where it can be scanned and imaged by electrons at various stages of its field evaporation, leading to a variety of imaging modes. During triggered field evaporation of the APT specimen, ions are collected on a position sensitive and time-resolved ion detector facing the specimen, leading to 3D reconstruction atom by atom.

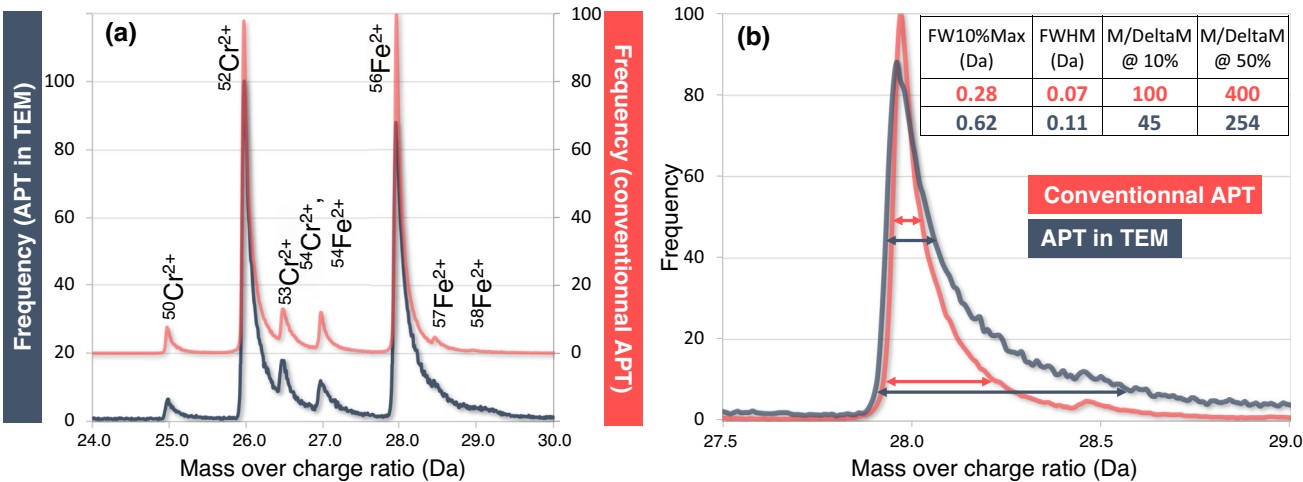

**Fig. 3 | Compared mass spectra of the of the same material analysed by APT in TEM or with a conventional APT (LEAP 5000 XS®). a** Representation in the range of detected Fe and Cr isotopes. **b** Enlarged view of mass spectra at the position of $^{56}Fe^{2+}$ peak used to evaluate the respective mass resolution of the APT-TEM setup and the LEAP 5000 XS. The table inserted in **b** allows to compare for each instrument, for the $^{56}Fe^{2+}$ peak, the full width (FW) at 10%, the full width at half maximum (FWHM) the mass over mass incertitude ratio (M/DeltaM).

mode is 0.23 nm, as specified by the supplier. Each picture of the sequence was captured with a Gatan US 1000 camera in TEM mode. At the magnification used, the pixel size is 0.24 × 0.24 nm2, which corresponds to the point-to-point resolution of the instrument. For the parameters of acquisition, an average electrostatic field ranging from 10 to 27 V/nm could be estimated at the tip apex from the measurement of the instantaneous tip radius. Such values are in the range of fields encountered in APT. Our results hence show that no obvious distortion of the TEM image of the tip apex is evidenced despite such a high electrostatic field. This demonstrates the possibility to follow dynamically the geometrical evolution of an APT specimen during its field evaporation.

With the demonstration that APT acquisitions can be carried out in standard (S)TEMs, the completion of this instrumental work will enable researchers to carry out original microscopy experiments. These allow the integration of multiple datasets across various dimensions (such as diffraction, electrostatic field, composition fields), along with the temporal aspect inherent in sample evaporation dynamics. This integration promises a deeper comprehension of the mechanisms governing nanoscale matter and future developments underway in connection with this project or the TOMO project[47] will soon enable experiments to be carried out with improved configurations (including improved vacuum, higher detection angle and laser pulses). For many topics of materials sciences, bringing APT to TEMs represents a rare opportunity to link structure, chemistry and properties in complex 3D materials architectures. Due to APT's unique capability in 3D mapping of the nanoscale distribution of the lightest elements, its convergence with STEM is poised to significantly enhance understanding in pivotal research domains, including the sustainability of Li-based batteries and the phenomenon of hydrogen embrittlement in metals.

## Methods
### APT setup and STEM observations
APT-TEM analyses were performed in a JEOL F2 microscope operating at 200 kV equipped with a dry pumping system. Vacuum level during APT analyses was $8 \times 10^{-6}$ Pa. The APT detection system mounted on the JEOL F2 is an aDLD (advanced delay-line detector) detection system[48]. The APT detector setup has been designed to be mounted on the (S)TEM column to fit the objective lens (OL) apertures port. This port is directly located in front of the specimen holder. The high

contrast aperture located bellow the TEM pole piece is a good alternative to the OL apertures, which were hence removed. The standard JEOL flange used in this port has been modified to be compatible with standard DN-CF flange used in APT development. An optimised vacuum chamber and a specific flange have been designed to set a conventional APT detection system. The system is made off a Hamamatsu MCP assembly coupled with a RoentDek 2 d delay-line detector (DLD). The detection system has an effective diameter of 42 mm and a detection efficiency of 50%. It is placed on the optical axis of the specimen at a distance of 470 mm. The detection angle is then close to ±2.5° that gives a physical angle close to ±4°. Acquisition were performed at 78 K, with a pulse fraction of 20 % and a pulse repetition rate of 20 kHz.

The APT-(S)TEM holder is composed of 4 elementary parts that allow applying a DC Voltage up to 8 kV coupled with voltage pulses up to 2 kV at repetition rate of 20 kHz (see picture in Figure S4 of the supplementary information). The holder can be cooled down to 78 K within vacuum facilities. The main body of the APT-(S)TEM holder has been designed according the double O-ring JEOL design to fulfil a complete compatibility with JEOL TEM F2 goniometer. This body is connected to a specific nitrogen circulation circuit through a vacuum chamber composed of two connections used to apply DC and pulse voltages and to measure the sample temperature. A specific ceramic piece, made of alumina, ensures the specimen assembly, the adapted connection to the nitrogen circulation and the insulation from the link used to apply high voltages.

A complete check-up of the radioprotection measurement shows no X-Ray leakage at the holder or the detector position.

### Compared APT analysis using the LEAP 5000 XS®
Temperature was set to 78 K, the fraction of pulse voltage and the pulse repetition rate was set to 25 kHZ. These settings allow comparing directly the effect of the TEM environment on the quantitativity of data, therefore avoiding any effect of temperature or pulse repetition frequency on the mass spectrum.

### Material
The material shown in Fig. 1 is an Al-1.7at%Ag alloy wherein a fine precipitation of Ag-rich precipitates was by annealing the material for 3 h at 150 °C, after a solution heat treatment applied at 560 °C for 5 h and followed by a water quench.

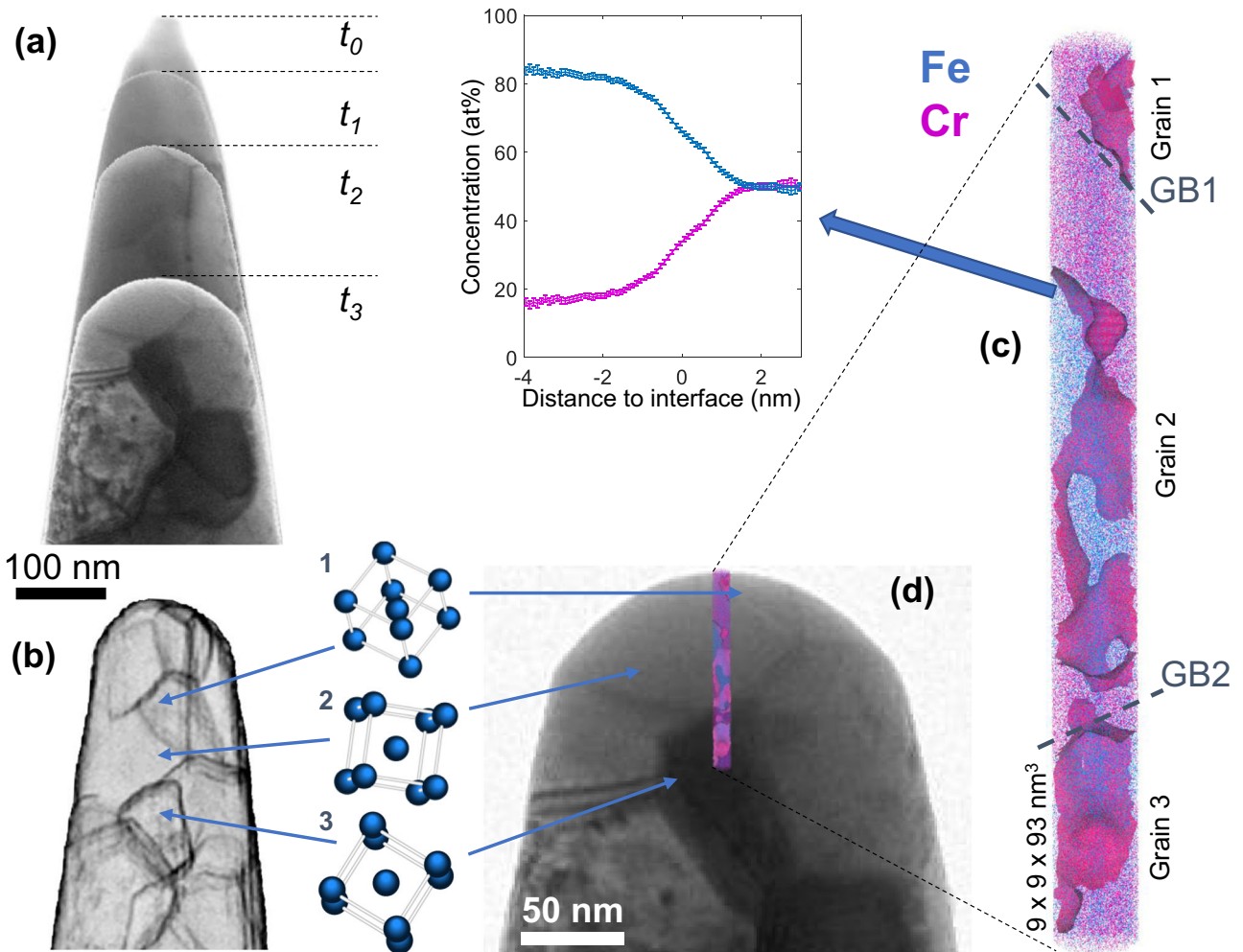

**Fig. 4 | In situ correlative analysis by STEM and APT of a Fe-51.4at%Cr alloy with ultrafine grain structure. a** STEM bright field images of the APT specimen recorded in between successive APT acquisitions (steps $t_0$ to $t_3$). **b** Map of grain boundaries calculated at time $t_2$ using the ACOM TEM methodology[50] (see additional data in the supplementary information). The crystalline orientation of three grains is displayed. **c** APT reconstruction at step $t_3$, with an isosurface of 28at% Cr composition. A composition profile calculated across a selected isosurface is displayed with the standard deviation. **d** Overlap of the APT reconstruction with the BF STEM image acquired at step t3, wherein the respective locations of the 3 grain orientations crossed during the APT analysis are indicated.

The Fe-51.4Cr (at%) used in this study was previously investigated in a coarse grain structure[54]. In order to reach an ultrafine grain size distribution, severe plastic deformation was applied by high-pression torsion (HPT) before an ageing heat treatment of 100 h at 525 °C. The HPT process was performed under 6 GPa with a deformation rate of 1 revolution per minute and a total of 10 revolutions at room temperature.

## Specimen preparation

APT needles were prepared by commonly used lift-out procedure on Thermo Scientific Helios5UX dual beam scanning electron microscope/focused ion beam (SEM/FIB) setup[35]. To obtain a flat surface desired for APT lift-out, Fe-Cr sample was mechanically polished by diamond abrasive solutions with grain size as low as 0.25 micrometres. The lifted-out wall was transferred to W pre-tips and glued by Pt deposition. W pre-tips were prepared from wires (99.99% W) by electropolishing using solution of 3% NaOH in water and voltage ranging from 5–10 V[33]. Generally, from one FIB lift-out, four APT tips were prepared.

The tip sharpening was performed using a set of circular masks by constantly reducing the inner diameter from two micrometres to 250 nanometres. The goal was to obtain a sharp needle with a diameter of below 100 nm at the tip apex. For all steps accelerating voltage of 30 kV was utilised, whilst ion currents were decreasing from 0.44 nA to 90 pA. Afterwards, a final cleaning procedure with low energy ion beam (2 kV, 100 pA) was performed to remove the top 200 nm of Ga-contaminated layer.

## Reporting summary

Further information on research design is available in the Nature Portfolio Reporting Summary linked to this article.

## Data availability

The following data have been made accessible under the CC BY 4.0 license. The raw data of the 4D-STEM acquisition presented in Fig. 4 has been deposited in the Zenodo database under the https://doi.org/10.5281/zenodo.13881947 (https://zenodo.org/records/13881947). The APT data set presented in Fig. 4 has been deposited in the Zenodo database under the https://doi.org/10.5281/zenodo.13881947 (https://zenodo.org/records/13886599). The mass spectrum data presented in Fig. 3 has been deposited in the Zenodo database under the https://doi.org/10.5281/zenodo.13881822 (https://zenodo.org/records/13881822).

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

## Acknowledgements

W.L. thanks Region Normandie for its financial support through the RIN Tremplin FusionSATMET project. W.L. acknowledges ANR for its support through the project SpinodalDesign (ANR-22-CE08-0016-01). We are grateful to Dr. Frédéric Danoix for providing the cast Fe-Cr alloy used in this work. Dr. Edgar Rauch is thanked for his help and advices about the ASTAR methodology. Dr. Xavier Sauvage is thanked for his helpful comments on this paper. JEOL company and its engineers are acknowledged for allowing the team at GPM to perform this instrumental development on the JEOL F200 microscope.

## Author contributions

G.D.C. designed the APT detector setup. C.C. supervised the connection of the APT to the STEM, performed experiments and data mining. A.N. designed most of the APT-TEM holder. C.V. realised some of the mechanical parts used for implementing the APT on the STEM, including parts of the APT-TEM holder. A.Z. performed specimen preparations in FIB SEM. J.M. realised heat treatments and specimen preparation of the Fe-Cr alloy. M.I. performed data mining with ASTAR software. K.E. realised the high-pressure torsion on the Fe-Cr alloy. F.V. performed data mining and interpretation and co-supervised this work. W.L. developed the research concepts, supervised this work, performed experiments, data mining and interpreted the data. W.L. and F.V. wrote the manuscript.

## Competing interests

The authors declare no competing interests.
