## [Transparent Peer Review file · Nature Communications]

Bringing Atom Probe Tomography to Transmission Electron Microscopes

Corresponding Author: Professor Williams Lefebvre

Version 0:

Reviewer comments:

Reviewer #2

(Remarks to the Author)

The manuscript titled “Bringing Atom Probe Tomography to Transmission Electron Microscopes” has been reviewed. In this work, the authors describe combining the two high spatial resolution imaging techniques of Atom Probe Tomography and Transmission Electron Microscopy into one single instrument – specifically the addition of APT to an existing TEM instrument. The research falls on the back of previous published efforts and expertise of the group who have described the design and fabrication of custom lab-made APT instrumentation, and it is this instrumentation that has been added to an existing TEM instrument. The authors present first results from this effort, showing APT and TEM results obtained simultaneously within the same instrument. Although the concept of combining the two techniques into a single instrument is not new as the authors describe, the presented results are significant as they represent the first experimental demonstration of this technology showing both APT and TEM data.

Before this manuscript should be considered for publication, the authors should explicitly address the following comments.

1. There is a lack of critical details around:

a. the APT-(S)TEM holder. Beyond the simple sketch in Fig 2, there is no substantive description and associated figure that allows the community to assess the design and implementation. Please add such information.

b. the APT detector system. Beyond just referencing a previous description of a custom APT detector technology (Ref. 43), there are no meaningful details. What specific design features were implemented or had to be overcome (e.g. how does the external field of the APT quantitatively affect (S)TEM resolution?)

2. Please add the following reference to the abstract (regarding 14-16) which helps to broaden the scope of interest, especially around efforts with SEM: Ceguerra, A. V.; Breen, A. J.; Cairney, J. M.; Ringer, S. P.; Gorman, B. P. Integrative Atom Probe Tomography Using Scanning Transmission Electron Microscopy-Centric Atom Placement as a Step Toward Atomic-Scale Tomography. *Microscopy and Microanalysis* 2021, 27 (1), 140–148. <https://doi.org/10.1017/S1431927620024873>.

3. To be more explicit for the reader, please add “in a single instrument” to the sentence on page 2 so that it reads “..., which highlight the allure for combining (S)TEM with APT in a single instrument for high-resolution material analysis.”

4. Page 2: change word “rebuilt” to reconstructed to be consistent with terminology of the field.

5. Page 2: add “before, during, and after analysis” to the following sentence to drive the point home “Volumes are later rebuilt, atom-by-atom, according to geometrical models which mostly require prior knowledge (or hypothesis) of the specimen geometry, before, during, and/or after analysis.”

6. Reference the work Qi, J.; Oberdorfer, C.; Windl, W.; Marquis, E. A. Ab Initio Simulation of Field Evaporation. *Phys. Rev. Mater.* 2022, 6 (9), 093602. <https://doi.org/10.1103/PhysRevMaterials>. when describing:

“Indeed, the atom-by-atom field evaporation induces a dynamic evolution of the electrostatic environment at the vicinity of the specimen’s apex, which results in a dynamic change of geometrical parameters controlling the 3D reconstruction (see the evaporation sequence provided in the supplementary information).”

Reviewer #3

(Remarks to the Author)

Reviewers Report: Bringing Atom Probe Tomography to Transmission Electron Microscopes

By Da Costa et al

The correlation of experimental techniques is a critical research area, as the results achieved by two or more techniques are multiplicative, rather than additive. One of the holy grails of experimental correlation nanoscience is that of (S)TEM imaging and atom probe tomography; the chemical uncertainty of the latter and the geometrical imprecision of the latter can be resolved by correlation. A small number of combination studies have shown this, in "ex-situ" mode as the authors put it. Most in 2D/3D (TEM/APT) and an even smaller number in 3D/3D. The construction of a combined instrument, and in doing so overcoming the multiple challenges in their compatibility, in an area of world wide research. The authors show the impressive results of the first instrument of this kind, and while lacking the cutting edge of both techniques, the application to largely 'off the shelf' TEM instrumentation is to be applauded. The paper shows some excellent first results and is deserving of publication. There are a few minor additions and changes that should be made to aid clarity and to solve some oversights in referencing. Both otherwise this should be published as soon as possible given the effort from other researchers in this area.

Changes and additions –

1. Figures. These are very lacking in annotation and labelling. And need work on clarity and making the most of their results.

Figure 2 has almost no labels. For a complicated 3D diagram, this is mostly clear to an expert, but a more general reader would not know what they are looking at. Please label all elements, and make it clear that the solid angle cone from the needle to the detector is not a solid object.

Figure 3 For a paper about correlation it is not clear how the APT and TEM images correlate! There is no 'overlap figure' as we would expect from most of the prior studies. The STEM images on the left are very 'muddy' and unclear. Can some adjustment of these images be done to make the microstructural details more clear? Element d) is meant to show the comparison between in-situ APT in the new instrument and that of a conventional system. However this figure is far too small to show this. This is a critical result. Figure S(5) from the supplemental is much better here, concentrating on a single peak and showing the differences between the two methods. Ideally 3(d) should be a standalone figure.

2. References. Given the very small number of successful correlative studies, the authors should make sure they don't miss any. In particular the very early work of Blavette et al (Acta Mat 44, 1996), the excellent correlative work of Gault et al (Ultramicroscopy 111, 2011) and one of the very few electron tomography / APT studies on the exact same material by Xiong et al (M&M 20, 2014)

Version 1:

Reviewer comments:

Reviewer #2

(Remarks to the Author)

Regarding response to Reviewer #2:

1. There is a lack of critical details around:

a. the APT-(S)TEM holder. Beyond the simple sketch in Fig 2, there is no substantive description **and associated figure** that allows the community to assess the design and implementation. Please add such information.

I thank the authors for adding written details describing the APT-(S)TEM holder to the Methods Section. However, there is still no associated figure showing a photo of the holder to help the community assess the design and implementation. With much of the technological success of this project relying on the specialized and complex APT-(S)TEM holder, it is concerning that the authors still DO NOT provide any details other than 126 written description. This needs to be addressed. Considering how figure panels S3b and S3C are not very helpful, there should be plenty of room.

b. the APT detector system. Beyond just referencing a previous description of a custom APT detector technology (Ref. 43), there are no meaningful details. What specific design features were implemented or had to be overcome (e.g. how does the external field of the APT quantitatively affect (S)TEM resolution?)

I thank the authors for adding written details around the APT detector setup and response around how the external field of the APT affects the STEM analysis. Specifically though, the authors provide a detailed response around the STEM resolution. This is very important information. However, it is confusing why the authors made a decision NOT to include this in the text, especially as they say "Our result show that no obvious distortion of the TEM image of the tip apex is evidenced despite such a high electrostatic field. This is, to our opinion, an important results insofar as we demonstrate a possibility to follow dynamically the geometrical evolution of an APT specimen during its field evaporation" At least this should be added in the supplement. Such analysis only helps to support claims and elevate impact. Please include the author response below into the main or supplementary text.

"The point to point resolution of the JEOL F200 in TEM mode is 0.23 nm, as specified by the supplier. Each picture of the

EvapFeCr_JEOLF2.avi movie was captured with a Gatan US 1000 camera in TEM mode. At the magnification used, the pixel size is 0.24 x 0.24 nm², which corresponds to the point-to-point resolution of the instrument. For the parameters of acquisition, an average electrostatic field ranging from 10 to 27 V/nm could be estimated at the tip apex from the measurement of the instantaneous tip radius. Such values are in the range of fields encountered in APT. Our result show that no obvious distortion of the TEM image of the tip apex is evidenced despite such a high electrostatic field. This is, to our opinion, an important results insofar as we demonstrate a possibility to follow dynamically the geometrical evolution of an APT specimen during its field evaporation."

Reviewer #3

(Remarks to the Author)

All noted issues have been addressed by the authors by alterations and additions. As such I am happy to suggest publication of the manuscript in its current form.

Response to the reviewers' comments

Bringing Atom Probe Tomography to Transmission Electron Microscopes

By Gerald Da Costa¹, Celia Castro¹, Antoine Normand¹, Charly Vaudolon¹, Aidar Zakirov¹, Juan Macchi¹, Mohammed Ilhami¹, Kaveh Edalati², François Vurpillot¹, Williams Lefebvre¹

The authors are very grateful to the reviewers for their consideration of this work and their very helpful comments and suggestions which allowed improving the presentation of our results. Our point-by-point answers to the reviewers' comments is shown below in blue

Reviewer #2 (Remarks to the Author):

The manuscript titled “Bringing Atom Probe Tomography to Transmission Electron Microscopes” has been reviewed. In this work, the authors describe combining the two high spatial resolution imaging techniques of Atom Probe Tomography and Transmission Electron Microscopy into one single instrument – specifically the addition of APT to an existing TEM instrument. The research falls on the back of previous published efforts and expertise of the group who have described the design and fabrication of custom lab-made APT instrumentation, and it is this instrumentation that has been added to an existing TEM instrument. The authors present first results from this effort, showing APT and TEM results obtained simultaneously within the same instrument. Although the concept of combining the two techniques into a single instrument is not new as the authors describe, the presented results are significant as they represent the first experimental demonstration of this technology showing both APT and TEM data.

Before this manuscript should be considered for publication, the authors should explicitly address the following comments.

1. There is a lack of critical details around:

a. the APT-(S)TEM holder. Beyond the simple sketch in Fig 2, there is no substantive description and associated figure that allows the community to assess the design and implementation. Please add such information.

The following additional information about the APT-(S)TEM holder's design has been added in the Methods section.

The APT-(S)TEM holder is composed of 4 elementary parts that allow to apply DC Voltage up to 8kV coupled with voltage pulses up to 2kV at frequency of 20kHz. The holder can be cooled down to 78K within vacuum facilities. The main body of the APT-(S)TEM holder has been designed according the double O-ring JEOL design to fulfill a complete compatibility with JEOL TEM F200 goniometer. This body is connected to a specific nitrogen circulation circuit through a vacuum chamber composed of 2 connections used to apply DC and pulse voltages and to measure the sample temperature. A specific ceramic piece, made of alumina, ensures the specimen assembly, the adapted connection to the nitrogen circulation and the insulation from the link used to apply high voltages.

b. the APT detector system. Beyond just referencing a previous description of a custom APT detector technology (Ref. 43), there are no meaningful details. What specific design features were implemented or had to be overcome (e.g. how does the external field of the APT quantitatively affect (S)TEM resolution?

In order to answer the question about the specific design features raised by the reviewer and to improve the description of our setup, the following detailed information about the APT detector mounted on the TEM column has been added in the Methods section.

The APT detector setup has been designed to be mounted on the (S)TEM column to fit the objective lens (OL) apertures port. This port is directly located in front of the specimen holder. The high contrast aperture located below the TEM pole piece is a good alternative to the OL apertures, which were hence removed. The standard JEOL flange used in this port, has been modified to be compatible with standard DN-CF flange used in APT development. An optimized vacuum chamber and a specific flange have been designed to set a conventional APT detection system. The system is made off a Hamamatsu MCP assembly (42mm effective diameter) coupled with a RoentDek 2d delay line detector (DLD). A complete checkup of the radioprotection measurement shows no X-Ray leakage at the holder or the detector position.

As for the specific example raised by the reviewer concerning the effect of the external field generated by the APT on the actual resolution of the TEM, attention can be paid to the movie **EvapFeCr_JEOLF2.avi** provided as additional support of our results and for which snapshots are displayed in the supplementary information.

The point to point resolution of the JEOL F200 in TEM mode is 0.23 nm, as specified by the supplier. Each picture of the **EvapFeCr_JEOLF2.avi** movie was captured with a Gatan US 1000 camera in TEM mode. At the magnification used, the pixel size is 0.24 x 0.24 nm², which corresponds to the point-to-point resolution of the instrument. For the parameters of acquisition, an average electrostatic field ranging from 10 to 27 V/nm could be estimated at the tip apex from the measurement of the instantaneous tip radius. Such values are in the range of fields encountered in APT. Our result show that no obvious distortion of the TEM image of the tip apex is evidenced despite such a high electrostatic field. This is, to our opinion, an important results insofar as we demonstrate a possibility to follow dynamically the geometrical evolution of an APT specimen during its field evaporation.

2. Please add the following reference to the abstract (regarding 14-16) which helps to broaden the scope of interest, especially around efforts with SEM: Ceguerra, A. V.; Breen, A. J.; Cairney, J. M.; Ringer, S. P.; Gorman, B. P. Integrative Atom Probe Tomography Using Scanning Transmission Electron Microscopy-Centric Atom Placement as a Step Toward Atomic-Scale Tomography. *Microscopy and Microanalysis* 2021, 27 (1), 140–148. <https://doi.org/10.1017/S1431927620024873>.

This reference has been inserted in the introduction, as suggested by the reviewer and we are very thankful for this suggestion.

3. To be more explicit for the reader, please add “in a single instrument” to the sentence on page to so that it reads “..., which highlight the allure for combining (S)TEM with APT in a single instrument for high-resolution material analysis.”

The sentence has been modified accordingly.

4. Page 2: change word “rebuilt” to reconstructed to be consistent with terminology of the field.

This change has been done.

5. Page 2: add “before, during , and after analysis” to the following sentence to drive the point home “Volumes are later rebuilt, atom-by-atom, according to geometrical models which mostly require prior knowledge (or hypothesis) of the specimen geometry, before, during, and/or after analysis.”

The sentence has been modified accordingly.

6. Reference the work Qi, J.; Oberdorfer, C.; Windl, W.; Marquis, E. A. Ab Initio Simulation of Field Evaporation. Phys. Rev. Mater. 2022, 6 (9), 093602. <https://doi.org/10.1103/PhysRevMaterials>. when describing:

“Indeed, the atom-by-atom field evaporation induces a dynamic evolution of the electrostatic environment at the vicinity of the specimen’s apex, which results in a dynamic change of geometrical parameters controlling the 3D reconstruction (see the evaporation sequence provided in the supplementary information).”

This new reference has been inserted in the sentence identified by the reviewer. We are grateful to the reviewer to have suggested the use of this reference, which will be very useful to readers interested in this specific topic.

Reviewer #3 (Remarks to the Author):

Reviewers Report: Bringing Atom Probe Tomography to Transmission Electron Microscopes

By Da Costa et al

The correlation of experimental techniques is a critical research area, as the results achieved by two or more techniques are multiplicative, rather than additive. One of the holy grails of experimental correlation nanoscience is that of (S)TEM imaging and atom probe tomography; the chemical uncertainty of the latter and the geometrical imprecision of the latter can be resolved by correlation. A small number of combination studies have shown this, in “ex-situ” mode as the authors put it. Most in 2D/3D (TEM/APT) and an even smaller number in 3D/3D. The construction of a combined instrument, and in doing so overcoming the multiple challenges in their compatibility, in an area of world wide research. The authors show the impressive results of the first instrument of this kind, and while lacking the cutting edge of both techniques, the application to largely ‘off the shelf’ TEM instrumentation is to be applauded. The paper shows some excellent first results and is deserving of publication. There are a few minor additions and changes that should be made to aid clarity and to solve some oversights in referencing. Both otherwise this should be published as soon as possible given the effort from other researchers in this area.

All co-authors are pleased to read these general comments about their work, which acknowledge and encourage their efforts towards the success of this long-term instrumental project. Detailed answers and comments about the modifications done are provided below.

Changes and additions –

1. Figures. These are very lacking in annotation and labelling. And need work on clarity and making the most of their results.

Figure 2 has almost no labels. For a complicated 3D diagram, this is mostly clear to an expert, but a more general reader would not know what they are looking at. Please label all elements, and make it clear that the solid angle cone from the needle to the detector is not a solid object.

All necessary labels have been added to Figure 2 to clarify all the key features of the setup together with the available microscopy modes offered by this new configuration. In addition, we have indicated above the solid angle “Angle of projection for APT”, so that the reader would no interpret this cone as a hardware setup. We thank the reviewer 3 for these remarks.

Figure 3 For a paper about correlation it is not clear how the APT and TEM images correlate! There is no ‘overlap figure’ as we would expect from most of the prior studies. The STEM images on the left are very ‘muddy’ and unclear. Can some adjustment of these images be done to make the microstructural details more clear? Element d) is meant to show the comparison between in-situ APT in the new instrument and that of a conventional system. However this figure is far too small to show this. This is a critical result. Figure S(5) from the supplemental is much better here, concentrating on a single peak and showing the differences between the two methods. Ideally 3(d) should be a standalone figure.

Following the reviewer's comments, we have modified the Figure 3 (which has become Figure 4). Here are the changes made:

- The contrast of STEM images on the left has been enhanced to better image the crystallographic features. It must be pointed out that these images were recorded and are used at a rather low magnification, on purpose, to follow the evolution of the whole specimen after several APT acquisitions. Hence, the best resolution is not pursued here. Nevertheless, the reconstruction of grain boundaries at step 3 is still displayed, which allow identifying these crystallographic features which were of particular interest in this work.
- An overlap figure is provided, which replaces Figure 3 (d). The APT volume is superimposed to the BF STEM image so as to fit the exact location of the volume investigated.
- The mass spectrum of the previous Figure 3(d) has been removed and a new Figure 3 has been inserted. Originally, we thought that moving the details about the mass spectrum resolution to supplementary information was a preferable, to ease the presentation of our work to non-specialists. However, we perfectly agree on the fact that these mass spectra are one of the key results of this work. Following these comments, we have built a new Figure 3 using the previous Figure 3d and Figure S(5). This way, the reader accesses all details about the compared mass spectrum resolutions in the core of the article. In addition, we have inserted some sentences for it description in the text:

It must be mentioned that the mass spectrum data displayed in **Erreur ! Source du renvoi introuvable**. for the APT in JEOL F2 is plotted without any background subtraction. The mass resolution at 10% is more strongly affected for the APT inside the TEM than the Full Width at Half Maximum (FWHM) value. The FWHM value demonstrated here, without any hardware correction, allows an easy distinction of peaks in the present application.

Accordingly, details about the mass spectra have been removed from the supplementary information.

2. References. Given the very small number of successful correlative studies, the authors should make sure they don't miss any. In particular the very early work of Blavette et al (Acta Mat 44, 1996), the excellent correlative work of Gault et al (Ultramicroscopy 111, 2011) and one of the very few electron tomography / APT studies on the exact same material by Xiong et al (M&M 20, 2014)

These 3 references have been added in the introduction in the following sentence:

To extend the intrinsic capabilities of APT, correlative approaches combining (scanning) transmission electron microscopy (STEM and TEM), have established a cutting-edge methodology to perform microscopy at a near atomic scale⁹⁻¹⁶

Response to the reviewers' comments

Bringing Atom Probe Tomography to Transmission Electron Microscopes

By Gerald Da Costa¹, Celia Castro¹, Antoine Normand¹, Charly Vaudolon¹, Aidar Zakirov¹, Juan Macchi¹, Mohammed Ilhami¹, Kaveh Edalati², François Vurpillot¹, Williams Lefebvre¹

Once again, the authors are very grateful to the reviewers for their consideration of this work and their very helpful comments and suggestions. You will find below in blue our point-to-point response to reviewer#2's comments and suggestions.

REVIEWER COMMENTS

Reviewer #2 (Remarks to the Author):

Regarding response to Reviewer #2:

1. There is a lack of critical details around:

a. the APT-(S)TEM holder. Beyond the simple sketch in Fig 2, there is no substantive description **and associated figure** that allows the community to assess the design and implementation. Please add such information.

I thank the authors for adding written details describing the APT-(S)TEM holder to the Methods Section. However, there is still no associated figure showing a photo of the holder to help the community assess the design and implementation. With much of the technological success of this project relying on the specialized and complex APT-(S)TEM holder, it is concerning that the authors still DO NOT provide any details other than 126 written description. This needs to be addressed. Considering how figure panels S3b and S3C are not very helpful, there should be plenty of room.

Two photos of the holder have been added in Figure S4 of the supplementary information, recalling the information given in the methods section in the caption. Some more details are added below this figure to help the reader understanding the design.

The availability of these pictures has been mentioned in the Methods section.

b. the APT detector system. Beyond just referencing a previous description of a custom APT detector technology (Ref. 43), there are no meaningful details. What specific design features were implemented or had to be overcome (e.g. how does the external field of the APT quantitatively affect (S)TEM resolution)?

I thank the authors for adding written details around the APT detector setup and response around how the external field of the APT affects the STEM analysis. Specifically though, the authors provide a detailed response around the STEM resolution. This is very important

information. However, it is confusing why the authors made a decision NOT to include this in the text, especially as they say "Our result show that no obvious distortion of the TEM image of the tip apex is evidenced despite such a high electrostatic field. This is, to our opinion, an important results insofar as we demonstrate a possibility to follow dynamically the geometrical evolution of an APT specimen during its field evaporation" At least this should be added in the supplement. Such analysis only helps to support claims and elevate impact. Please include the author response below into the main or supplementary text.

"The point to point resolution of the JEOL F200 in TEM mode is 0.23 nm, as specified by the supplier. Each picture of the EvapFeCr_JEOLF2.avi movie was captured with a Gatan US 1000 camera in TEM mode. At the magnification used, the pixel size is 0.24 x 0.24 nm², which corresponds to the point-to-point resolution of the instrument. For the parameters of acquisition, an average electrostatic field ranging from 10 to 27 V/nm could be estimated at the tip apex from the measurement of the instantaneous tip radius. Such values are in the range of fields encountered in APT. Our result show that no obvious distortion of the TEM image of the tip apex is evidenced despite such a high electrostatic field. This is, to our opinion, an important results insofar as we demonstrate a possibility to follow dynamically the geometrical evolution of an APT specimen during its field evaporation."

Once again, the suggestion of Reviewer#2 really helps improving the impact of our work and we are grateful for this. We agree that the effect of electrostatic field on TEM resolution is a major concern about the feasibility of unifying TEM and APT in a single instrument and we agree that we did not enough insist on the fact that our work shows that dynamic imaging of field evaporation, for the purpose of APT reconstruction, is clearly possible.

Following the reviewer's suggestion, we have inserted the above paragraph in the core of the paper, with slight adjustments to ensure the consistency of the paper.

Reviewer #3 (Remarks to the Author):

All noted issues have been addressed by the authors by alterations and additions. As such I am happy to suggest publication of the manuscript in its current form.